# PD1 Deficiency Modifies Cardiac Immunity during Baseline Conditions and in Reperfused Acute Myocardial Infarction

**DOI:** 10.3390/ijms23147533

**Published:** 2022-07-07

**Authors:** Lars Michel, Sebastian Korste, Armin Spomer, Ulrike Barbara Hendgen-Cotta, Tienush Rassaf, Matthias Totzeck

**Affiliations:** Department of Cardiology and Vascular Medicine, West German Heart and Vascular Center, University Hospital Essen, Hufelandstraße 55, 45147 Essen, Germany; lars.michel@uk-essen.de (L.M.); sebastian.korste@uk-essen.de (S.K.); armin.spomer@uk-essen.de (A.S.); ulrike.hendgen-cotta@uk-essen.de (U.B.H.-C.); tienush.rassaf@uk-essen.de (T.R.)

**Keywords:** cardio-oncology, immune checkpoint, immunology, ischemia/reperfusion injury, reperfused acute myocardial infarction, programmed cell death protein 1

## Abstract

The programmed cell death protein 1 (PD1) immune checkpoint prevents inflammatory tissue damage by inhibiting immune reactions. Understanding the relevance of cardiac PD1 signaling may provide new insights into the inflammatory events under baseline conditions and disease. Here, we demonstrate distinct immunological changes upon PD1 deficiency in healthy hearts and during reperfused acute myocardial infarction (repAMI). In PD1-deficient mice, upregulated inflammatory cytokines were identified under baseline conditions including cardiac interleukins and extracellular signal-related kinase 1/2 (ERK1/2). A murine in vivo repAMI model to determine inflammatory changes in the early phase showed downregulation of the ligand PDL1, paralleled by an endothelial injury, indicated by loss of the CD31 signal. Immunofluorescence imaging showed decreased PDL1 expression specifically in the infarct zone, highlighting an involvement in PDL1 in myocardial injury response. Pharmacological depletion of PD1 prior to repAMI did not alter the area of infarction but led to increased numbers of CD8^+^ T cells in treated mice. We conclude that PD1/PDL1 signaling plays a significant role in healthy hearts and repAMI, emphasizing the relevance of adaptive immunity during myocardial injury. The findings highlight the risk for adverse outcomes from acute myocardial infarction in the growing group of patients receiving immune checkpoint inhibitor therapy.

## 1. Introduction

Cardiovascular (CV) diseases are the leading causes of death worldwide [1]. The onset, progression, and healing of CV diseases are impacted by numerous factors, including personal risk factors, vascular function, metabolism, and inflammatory changes. Cardiac immunity plays a key role in maintaining functional integrity in healthy conditions and in mediating the response to various stressors, e.g., during an acute myocardial infarction and the postinfarction period [2,3]. The dense capillary network of the myocardium forms a large contact area with the immune system. Cardiac tissue is therefore particularly vulnerable to inflammatory reactions, and maintaining cardiac immune quiescence during healthy conditions is key to preventing detrimental inflammatory effects [4]. Inflammation itself plays a key role in mediating the progression and recovery from different forms of myocardial injury, particularly myocardial infarction [2,3]. Myocardial infarction is induced by the occlusion of a blood vessel usually by arteriosclerotic plaque rupture and thrombus formation. While myocardial ischemia induces the first phase of cardiac injury, reperfusion itself represents a second phase of cardiac injury, leading to myocardial inflammation. Reperfusion-related effects comprise up to 50% of cardiac injury from acute myocardial infarction (repAMI), particularly mediated by myocardial inflammation, which is orchestrated by a cascade of infiltrating immune cells that are tightly regulated by pro-and anti-inflammatory signaling pathways, including immune checkpoints [2,3].

Immune checkpoints are a group of inhibitory receptors that mediate immune activity to counteract exaggerated inflammatory reactions or autoimmune diseases [5]. Immune checkpoints have recently become popular as the target of immune checkpoint inhibitor (ICI) therapy, which induces an anti-tumor immune reaction by a blockade of immune checkpoints. Inhibitors of the programmed cell death protein 1 (PD1) and its ligand PDL1 are most commonly applied [6]. Anti-PD1/PDL1 therapy has improved the treatment of various advanced cancers including melanoma, non-small-cell lung cancer, lymphoma, and renal cell carcinoma [5]. Due to its clinical success, ICI therapy is increasingly applied, and new substances are being developed. The growing application of ICI therapy was, however, associated with the recognition of immune-related cardiovascular adverse events, including ICI-related myocarditis, arrhythmia, left ventricular dysfunction, takotsubo syndrome, and accelerated development of coronary artery disease and acute coronary syndrome [7,8,9]. 

ICI therapy was shown to induce deleterious CV effects as seen in clinical data and preclinical models, which was attributed to various mechanisms. While a myocardial infiltration with different T-cell subpopulations is generally considered the main mechanism of acute ICI-related myocarditis [10], various effects on myocardial integrity have been observed, including metabolic and functional dysregulation as recently shown [11]. Furthermore, evidence from myocarditis mouse models indicates a more general involvement of PD1/PDL1 signaling in myocardial inflammation [12]. It has also been suggested that PD1 and PDL1 are involved in the inflammatory processes following myocardial ischemia. However, in vivo evidence is lacking [13]. In sum, the available body of evidence indicates that PDL1, which is highly expressed in cardiac endothelial cells, shields the heart from immune reactions during baseline conditions and myocardial injury [4]. Understanding specific effects of cardiac PD/PDL1 immune checkpoint signaling during healthy conditions and cardiac injury may help us to identify specific mechanisms that lead to adverse outcomes, eventually leading to a better understanding of the effects of ICI therapy in the growing number of affected patients. This study aims to assess the baseline effects of cardiac PD1/PDL1 signaling and its response to myocardial repAMI.

## 2. Results

### 2.1. Cardiac PD1 Deficiency during Baseline Conditions

The relevance of PD1/PDL1 for cardiac immunity during baseline conditions is unclear. To determine potential inflammatory changes during baseline conditions, key inflammatory cytokines were assessed in whole heart-tissue homogenates of C57BL/6J wild-type (wt) and C57BL/6J *Pdcd1*^−/−^ mice. The analysis determined increased expression of the proinflammatory cytokine interleukin 1α (IL1α) in *Pdcd1^−/−^* mice (fold change 1.61; *p* = 0.0286; Figure 1A). Additionally, IL4, which induces the differentiation of T cells, showed a tendency towards elevated levels in *Pdcd1*^−/−^ mice compared to wt mice without reaching statistical significance (fold change 3.325; *p* = 0.1000; Figure 1B). Extracellular signal-related kinases 1/2 (ERK1/2) are protein–serine/threonine kinases that are involved in key inflammatory processes including adhesion, proliferation, and differentiation as part of the RAS-RAF-MEK-ERK signal transduction cascade. Phosphorylated ERK1/2 was significantly upregulated in cardiac tissue of *Pdcd1*^−/−^ mice (fold change 2.231; *p* = 0.0286; Figure 1C). Oxidative stress is a hallmark of cardiac injury and endothelial dysfunction [14]. To determine changes in redox signaling, the expression of NADPH oxidase 1 (NOX1) was assessed. No difference was observed, indicating no sign of involvement of redox signaling in PD1-related effects during baseline conditions (Figure 1D).

### 2.2. PDL1 Expression during repAMI 

PDL1 is highly expressed in cardiac endothelial cells [11,15]. To test whether cardiac expression of PDL1 is altered during repAMI, C57BL/6J wt mice were subjected to in vivo repAMI with 45 min ischemia followed by 24 h or 72 h of reperfusion. The expression of PDL1 in left ventricular tissue was assessed by flow cytometry, preceded by the identification of cardiac cell subtypes (endothelial cells, fibroblasts, cardiomyocytes, and leucocytes) as described previously (Figure 2A) [11]. As expected, the expression of CD31 decreased following repAMI as a sign of endothelial injury/dysfunction. In parallel, the expression of PDL1 on CD31^+^ endothelial cells was found significantly decreased at 72 h following repAMI (28.4% decrease at 72 h; *p* = 0.0002; Figure 2B,C). As evidenced before, cardiomyocytes and fibroblasts did not show significant expression of PDL1 [11]. The spatial distribution of PDL1 in infarcted myocardium was determined by immunofluorescence staining. After 24 h of repAMI, we determined that PDL1 was downregulated in the infarcted area, visualized by a loss of CD31 expression, and loss of cardiac troponin T. PDL1 expression was not changed in the remote zone (Figure 3).

### 2.3. PD1-Depletion during repAMI 

Anti-PD1 ICI therapy has been associated with various aspects of coronary artery disease, including adverse outcomes in acute coronary syndrome, accelerated progression, and accelerated development in atherosclerosis [7,16,17]. To determine a potential impact on the infarct size during PD1-blockade, mice were treated with anti-PD1 therapy or IgG2A control starting at day −8 every other day until induction of repAMI (Figure 4A). At 24 h, infarct sizes did not differ between animals receiving anti-PD1 therapy or IgG2A control, indicating no impact on infarct size in the acute state (Figure 4B–D).

As PD1/PDL1 signaling mainly affects lymphocyte migration, differentiation, and activity, we assessed infiltrating immune cells at 72 h after repAMI by applying a specific immune cell panel as established previously [11] (Figure 5A). The analysis revealed increased CD8^+^ T cells in mice receiving anti-PD1 therapy compared to those receiving the IgG2A control (33.6% increase; *p* = 0.0286). The number of B cells and the number of CD4^+^ T cells were not significantly altered (Figure 5B,C).

To determine whether the observed immunological changes already have an impact on cardiac function in the early phase, we conducted mouse echocardiography. No significant differences were observed regarding cardiac dimensions, left ventricular muscle mass, and left ventricular systolic function (Figure 6A–F). The assessment of high-sensitive cardiac troponin I did not reveal significant differences (Figure 6G).

## 3. Discussion

The present study provides evidence of the effects of the PD1/PDL1 immune checkpoint in healthy hearts and repAMI. The main findings are (i.) PD1 deficiency induces elevated levels of inflammatory cytokines in healthy heart tissue; (ii.) PDL1 is markedly downregulated in the infarct area following repAMI; (iii.) anti-PD1 therapy promotes myocardial T-cell infiltration during repAMI without affecting infarct sizes in the acute reperfusion phase.

The increasing application of ICI therapy in patients with advanced cancers has unmasked a variety of CV complications with significant effects on the morbidity and mortality of treated patients. In patients receiving anti-PD1 therapy, a 1-year cardiac event rate of up to 9.7% was reported, hence indicating an imminent medical need for understanding the mechanisms behind such complications [18,19]. While early evidence identified ICI-related myocarditis as the main phenomenon, recent evidence showed that ICI-related CV effects are more diverse, including effects on various immune cell fractions, myocardial energy metabolism, lipid distribution, mitochondrial function, and contractility [11]. The cardiac concentration of CD4^+^ and CD8^+^ T cells increases upon anti-PD1 therapy, with signs of increased T-cell activation [11]. In parallel, myocardial macrophage activity increases during anti-PD1 therapy, showing enhanced M1 polarization with elevated levels of IL1β, IL6, and inducible nitric oxide synthetase (iNOS) [20]. Distinct changes in cardiomyocyte key regulator proteins have been identified, particularly components of intracellular energy substrate handling and oxidative phosphorylation [11,21]. Blockade of PD1 was furthermore shown to aggravate a radiation-induced cardiac injury, leading to decreased cardiac function and pronounced myocardial fibrosis [22]. The role of tumor necrosis factor α (TNFα) has been identified independently in multiple studies, showing a central role in changes related to immune cells, radiation-related toxicity, and cardiac function [11,20,22]. 

The present data add another important finding on molecular mechanisms observed during PD1-deficiency. IL1α is a pleiotropic inflammatory cytokine that is produced by macrophages, monocytes, fibroblasts, dendritic cells, and endothelial cells. IL1 is involved in a broad range of inflammatory effects, including immune cell activation, acute phase reaction, and fever. Here, PD1 deficiency was associated with the upregulation of IL1 in cardiac tissue, thus suggesting a ubiquitous proinflammatory role of IL1 in the cardiac inflammasome in the absence of PD1. Strikingly, IL1α is the key activator of TNFα, and both cytokines show synergistic effects in promoting tissue inflammation [23,24]. Mitogen-activated protein (MAP) kinases are substantial components in a variety of biological processes, including the regulation of differentiation, the release of growth factors, cell survival and apoptosis, and the induction of cytokines [25]. ERK1/2 is expressed in different cardiac cell types including cardiomyocytes, endothelial cells, and fibroblasts. It is particularly involved in cardiomyocyte homeostasis and cell death regulation, response to stress including repAMI, and cardiac remodeling [3]. Particularly, ERK1/2 has been linked to heart failure and pathological hypertrophy. ERK1/2 was 2.2-fold upregulated in *Pdcd1*^−/−^ mice, hence indicating a potential relevance of the RAS-RAF-MEK-ERK signal transduction cascade in PD1-deficiency. Strikingly, ERK1/2 promotes TNFα in macrophages via nuclear factor κB (NFκB), thereby indicating another connection of PD1-related effects to TNFα [26]. In a recent model for anti-PD1 ICI-related cardiotoxicity, the blockade of TNFα has been shown to mitigate cardiotoxic effects, thus underlining its key role in the pathogenesis of PD1/PDL1-related CV adverse effects.

Distinct effects of PD1/PDL1 in myocardial repAMI have been demonstrated in the present study, showing a pronounced decrease in PDL1 upon repAMI, while PD1-blockade induced an increased infiltration of CD8^+^ T cells. The effects of PD1/PDL1 in coronary artery disease and acute myocardial infarction are only beginning to be understood. In a prior study using an ex vivo rat model using the Langendorff apparatus to induce myocardial injury, increased expression of PDL1 was observed during reperfusion, which was recapitulated for cryo-injury as a second form of myocardial injury [13]. In the present study, different observations were made by showing decreased PDL1 expression in flow cytometry and immunofluorescence. While the reason for the diverging results remains unclear, profound differences in the experimental setups can be found, as the present project shows differences in PDL1 expression in an in vivo repAMI model, which is expected to show immunological changes more accurately than the ex vivo model in the absence of a humoral immune system [13]. A decreased expression of PDL1 in the infarcted area seems plausible to support the inflammatory reaction with subsequent remodeling and tissue healing, but a longer observation period is necessary to assess this conclusively. Despite the acute changes shown here, PD1 was furthermore shown to accelerate atherosclerosis by promoting plaque-related inflammation [15]. In preclinical models, PD1 deficiency induced larger lesions with increasing numbers of infiltrating macrophages and T cells, accompanied by increased levels of TNFα [17,27,28]. These findings were recapitulated in a series of patients showing increasing aortic plaque sizes upon ICI therapy [16]. Until now, it has been unclear how the acute effects regarding PDL1 expression and T-cell infiltration and the long-term effects showing accelerated atherosclerosis synergize, but a distinct effect can clearly be expected. Prospective patients’ data are now needed to further characterize the impact of short and long-term effects of ICI therapy on coronary artery disease and acute coronary syndrome.

### Limitations

As the present findings are based on single preclinical models, the translation of the results to the clinical situation may be limited. While distinct molecular and immunological findings could be demonstrated, robust endpoints including cardiac function, long-term cardiac remodeling, and survival were not assessed, hence requiring further research in future projects. As outlined before, the effects of ICI therapy on plaque progression were not assessed in this project, which may alter the results in affected patients. Finally, all present findings are highly experimental, and distinct consequences for clinical work cannot be derived so far. Future translation research will provide the basis for improved patient care.

## 4. Materials and Methods

C57BL/6N wt, C57BL/6J *Pdcd1*^−/−^ and C57BL/6J wt mice were purchased (strain code 0027, Charles River Laboratories, Wilmington, MA, USA; strain number #028276 and #000664, The Jackson Laboratory, Bar Arbor, ME, USA) and bred at the animal facility of the University Hospital Essen, Germany [29]. Experiments involving animals were conducted according to the “European Convention for the Protection of Vertebrate Animals used for Experimental and other Scientific Purposes” (EU Directive 2010/63/EU) and in accordance with institutional guidelines. Approval for all experiments involving animals was granted by the Landesamt für Natur, Umwelt und Verbraucherschutz Nordrhein-Westfalen (NRW), Germany. 

### 4.1. Ischemia/Reperfusion (I/R) Heart Injury

In vivo repAMI was induced by transient occlusion of the left coronary artery (LCA) for 45 min followed by reperfusion for 1–5 days as described previously [30,31]. C57BL/6J *Pdcd1*^−/−^ and C57BL/6J wild-type mice were anesthetized by intraperitoneal injection of ketamine (bela-pharm, Vechta, Germany) and xylazine (Ceva Tiergesundheit, Düsseldorf, Germany). Mice were intubated and ventilated with 40% O_2_ and 1.6–2.0 Vol.% isoflurane (Piramal, Mumbai, India). Analgesia was maintained with buprenorphine. Left-lateral thoracotomy was performed followed by pericardiotomy and exposure of the LCA. The LCA was ligated for 45 min while maintaining anesthesia, followed by reperfusion. The tissue was then closed with sutures before the end of the anesthesia. For experiments involving antibodies, mice were injected intraperitoneally with 250 µg anti-PD1 (RMP1-14) or 250 µg rat IgG2a (2A3, both BioXCell, West Lebanon, NH, USA) every other day starting on day 8 before surgery, followed by day 2 after surgery. Analgesia was maintained with buprenorphine 3 times a day. Mice were killed with an isoflurane overdose.

### 4.2. 2,3,5-Triphenyltetrazolium Chloride (TTC) Evan’s Blue Staining

TTC and Evan’s blue staining were used to quantify the myocardial repAMI size as described previously [30,31]. Hearts were removed and flushed blood-free with PBS (Sigma-Aldrich, St. Louis, MO, USA) via cannulation of the ascending aorta. The LCA was then re-occluded by suture followed by injection of 1 mL 1% Evan’s blue dye (Sigma-Aldrich, St. Louis, MO, USA) to visualize the area at risk. Then, 1–2 mm transversal cross-sections of the myocardium were created and incubated in 1% TTC in 0.0774 M Na_2_HPO_4_ (Roth, Karlsruhe, Germany) and 0.0226 M NaH_2_PO_4_ (Sigma-Aldrich, St. Louis, MO, USA) at 37 °C for 5 min. Images were obtained with an M80 microscope with an IC80 HD camera (Leica, Wetzlar, Germany) and analyzed by a blinded investigator using ImageJ software.

### 4.3. Echocardiography

Mouse echocardiography was conducted using a VisualSonics Vevo ultrasound system (VisualSonics, Toronto, ON, Canada). Mice were anesthetized with isoflurane and placed on a heated plate. Heart rate, respiratory rate, and body temperature were monitored continuously and maintained within pre-defined limits. Parasternal long-axis and parasternal short-axis views at basal, mid-ventricular, and apical levels were acquired as B-mode and M-mode. Analysis was conducted post hoc using the VevoLab 3.2 software.

### 4.4. Flow Cytometry

Flow cytometry was conducted using single-cell suspensions from murine left ventricular cardiac tissue. Hearts from mice under baseline conditions and after repAMI with different reperfusion times were used for experiments. Mouse hearts were flushed blood-free with PBS for 3 min via puncture of the left ventricular apex. The right ventricular myocardium and the left and right atria were removed, and the remaining left ventricular tissue was manually dissected followed by enzymatic digestion as described previously by our group in 450 U/mL collagenase I, 60 U/mL hyaluronidase, 125 U/mL collagenase XI, 60 U/mL DNAse, and 20 mM Hepes (all Sigma-Aldrich, St. Louis, MO, USA) in 500 µL PBS at 37 °C for 40 min on a thermal shaker (Eppendorf, Hamburg, Germany) [11]. After Fc blocking to avoid non-specific binding of antibodies, cells were stained with PerCP or AF700 CD45, BV421 CD31, and BV605 CD140A to identify cardiac cell populations. PD1 and PDL1 were assessed using APC PD1, and PE PDL1 in left ventricular tissue (all BioLegend, San Diego, CA, USA). For immune cell analysis, whole heart suspensions were stained with AF700 CD45, PE-Dazzle594 CD11b, PerCP-Cy5.5 Ly6G, APC Ly6C, BV421 F4/80, FITC CD3, PE-Cy7 CD4, APC-Cy7 CD8, BV605 CD19, PE CD44, and Zombie aqua (all BioLegend, San Diego, CA, USA). Cells were analyzed using a BD FACS Aria III (BD Biosciences, San Jose, CA, USA). Post hoc data analysis was performed using FlowJo (FlowJo, LLC/BD Biosciences, San Jose, CA, USA).

### 4.5. Western Blot and Plasma Analysis

Hearts were flushed blood-free and snap-frozen in liquid nitrogen. The following steps comprise the homogenization of the hearts in RIPA buffer and gyration for 60 min at 4 °C, to merge the homogenate with a protease and phosphatase inhibitor (Thermo Fisher Scientific, Waltham, MA, USA). Next, prepared homogenates were centrifuged to acquire supernatant for protein analysis. A DC protein assay was used to determine protein concentrations (Bio-Rad, Hercules, CA, USA).

Protein was separated on a Bolt 4–12% BIS-TRIS Plus gel and transferred onto a nitrocellulose membrane using the iBlot2 device (both Thermo Fisher Scientific, Waltham, MA, USA) at 21 °C for 7 min, followed by blocking with milk and washing. The membranes were then incubated at 4 °C overnight with a solution containing the corresponding primary antibody (anti-Interleukin-1-alpha (ab7632), mouse, 1:5000; anti-Interleukin 4 (ab84269), mouse, 1:1000; anti-NOX1 (ab131088), mouse, 1:2000; anti-pERK1/pERK2 (Cell Signaling #4370), mouse, 1:1000 and anti-GAPDH loading control (ab8245) (Abcam, Cambridge, UK, and Cell Signaling, Danvers, MA, USA), followed by covering the membrane with the solution containing the matching secondary antibody (goat anti-rabbit IgG (H + L), HRP (32460; goat anti-mouse IgG, HRP (31430), both 1:5000 Thermo Fisher Scientific, Waltham, MA, USA) for 1 h at 21 °C. GAPDH (anti-GAPDH (ab8245), 1:5000, Abcam, Cambridge, UK) was used as the loading control. Bound proteins of interest were visualized by the SuperSignal West Femto Maximum Sensitivity Substrate Kit (Thermo Fisher Scientific, Waltham, MA, USA) using an Amersham 600 Imager (GE Healthcare, Chicago, IL, USA). 

Plasma cardiac troponin I was assessed using an automated high-sensitive troponin I assay (Centaur, Siemens, Berlin, Germany).

### 4.6. Immunohistochemistry

Sections of 4 µm myocardial tissue were created from paraffin-embedded formalin-fixated samples. Sections were deparaffinized and rehydrated, followed by antigen retrieval by incubation at 95 °C in Antigen Retrieval Buffer pH 6.0 (Abcam, Cambridge, UK). Samples were then incubated in 3% H_2_O_2_ for 10 min, followed by blocking in TRIS-buffered saline with 0,1% (*v*/*v*) Tween-20 (NaCl, TRIS, and Tween-20 (Roth, Karlsruhe, Germany) with 5% (*v*/*v*) normal goat serum (Invitrogen, Carlsbad, CA, USA) for 1 h. Slices were incubated in blocking buffer with the primary antibodies at 4 °C overnight (PDL1 (ab213480), rabbit, 1:50, Abcam, Cambridge, UK; CD31 (DIA-310), rat, 1:20, Dianova, Hamburg, Germany; or cardiac troponin T (MA5-12960), mouse, 1:50, Invitrogen, Carlsbad, CA, USA) followed by staining with secondary antibodies at room temperature for 1 h (anti-rabbit AlexaFluor680, 1:200; anti-rat Cy3, 1:200; or anti-mouse AlexaFluor594, all Invitrogen, Carlsbad, CA, USA). Nuclei were stained with DAPI 1:5000 (1:5000 in TBS-T, Invitrogen, Carlsbad, CA, USA) followed by preservation in Prolong gold antifade reagent (Invitrogen, Carlsbad, CA, USA). Conventional hematoxylin and eosin stainings were created from slices from the same specimens. Images were acquired using an Olympus BX51 microscope (Olympus, Shinjuku, Japan) and processed using ImageJ software.

### 4.7. Statistical Analysis

The Mann–Whitney U test was used to analyze data with two groups. Data including more than two groups were analyzed by the Kruskal–Wallis test and Dunn’s multiple comparison *post hoc* test. *p* < 0.05 was considered significant. All analyses were conducted using GraphPad Prism 9 (GraphPad Software, San Diego, CA, USA). All underlying data are available from the corresponding author upon reasonable request.

## 5. Conclusions

The increasing reports of CV complications from ICI therapy indicate an imminent medical need for a better understanding of the role of immune checkpoints for CV disease. The present study identifies a distinct upregulation of proinflammatory cytokines during PD1-deficiency. During myocardial repAMI, a profound downregulation of PDL1 within the infarct area was observed, while PD1 depletion led to increasing T-cell infiltration in the myocardium following repAMI. How the observed effects affect long-term outcomes and myocardial repair following repAMI will be an important subject of further investigations. Understanding the mechanistic effects of immune checkpoint signaling in CV disease may eventually help to improve the treatment of patients experiencing CV complications from ICI therapy.

## Figures and Tables

**Figure 1 ijms-23-07533-f001:**
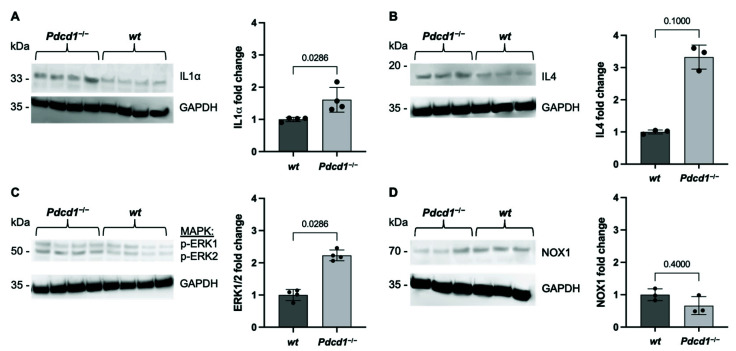
Elevated inflammatory cytokines in *Pdcd1*^−/−^ mice. Western blot analysis of whole heart homogenates from wt and *Pdcd1*^−/−^ mice showing (**A**) Interleukin 1α (IL1α) (*n* = 4 per group), (**B**) Interleukin 4 (IL4) (*n* = 3 per group), (**C**) phosphorylated extracellular signal-related kinase 1/2 (p-ERK1/2) (*n* = 4 per group), and (**D**) NADPH oxidase 1 (NOX1) expression (*n* = 3 per group) in *Pdcd1*^−/−^ mice compared to wt mice. Black dots indicate individual samples.

**Figure 2 ijms-23-07533-f002:**
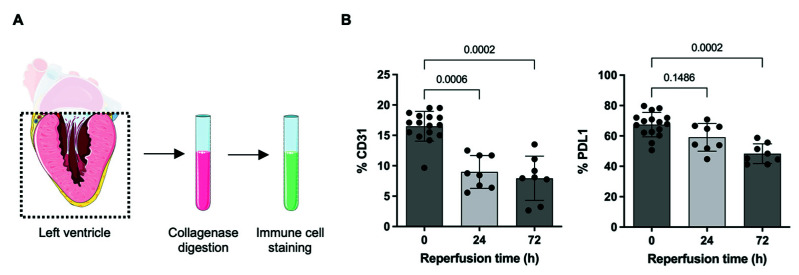
Decreased PDL1 expression following repAMI. (**A**) Scheme of flow cytometry-based analysis of immune cell analysis following collagenase digestion of left ventricular tissue. (**B**) Quantitative flow cytometry analysis showing decreased expression of the endothelial cell marker CD31 and decreased expression of PDL1 on endothelial cells following repAMI for 24 h or 72 h (*n* = 16 for baseline, *n* = 8 for 24 h reperfusion time, *n* = 8 for 72 h reperfusion time). Black dots indicate individual samples. (**C**) Exemplary flow cytometry data showing increasing CD45^+^ leucocytes and decreasing PDL1 expression upon repAMI.

**Figure 3 ijms-23-07533-f003:**
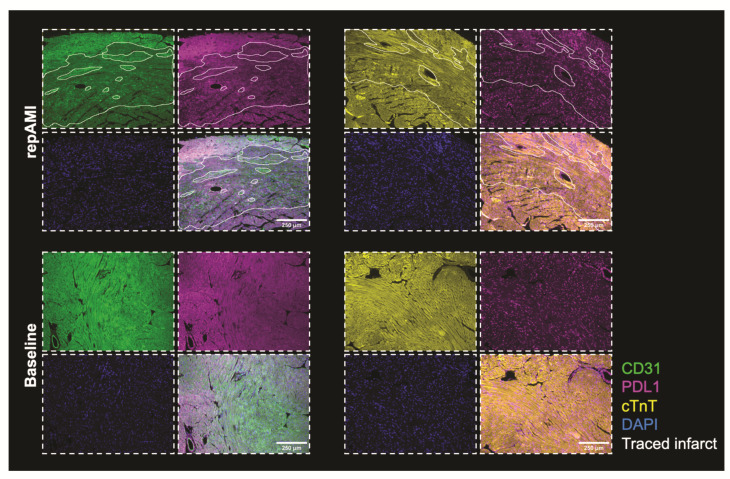
PDL1 distribution at baseline and following repAMI. Representative data from immunofluorescence imaging in mouse hearts 24 h following repAMI. PDL1 is shown in magenta, the endothelial cell marker CD31 in green, cardiac troponin T in yellow, and nuclei in blue. Imaging shows decreased PDL1 expression in the infarct zone which is identified by a decrease in CD31 expression and a decrease in cardiac troponin T.

**Figure 4 ijms-23-07533-f004:**
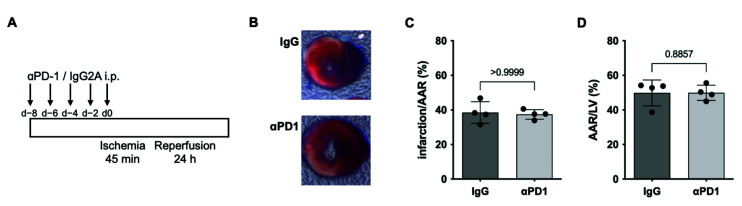
Infarct size in mice receiving αPD1 therapy. (**A**) Treatment scheme of mice receiving αPD1 therapy or IgG2A control followed by induction of repAMI with 24 h reperfusion. (**B**) Exemplary images and (**C**,**D**) quantification of infarct sizes determined by triphenyltetrazolium chloride (TTC) Evans’s blue staining (*n* = 4 per group). Black dots indicate individual samples.

**Figure 5 ijms-23-07533-f005:**
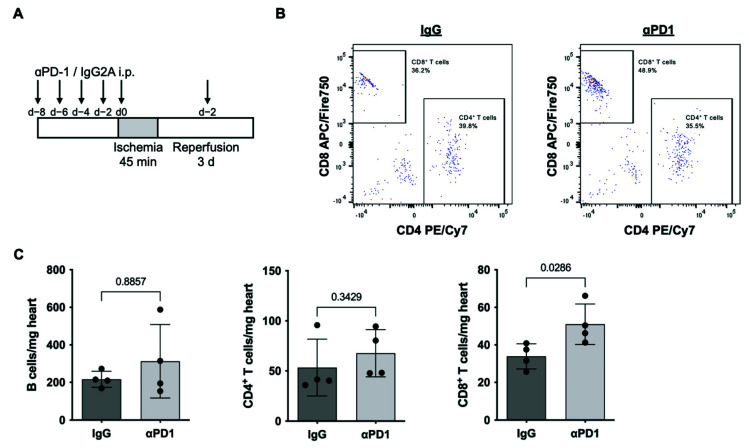
Increasing CD8^+^ T cells following repAMI in anti-PD1-treated mice. (**A**) Treatment scheme of mice receiving anti-PD1 therapy or IgG2A control followed by induction of repAMI with subsequent treatment and 72 h reperfusion. (**B**) Exemplary flow cytometry data showing CD3^+^ T cells stratified for CD4^+^ and CD8^+^. (**C**) Quantification of B cells, CD4^+^ T cells, and CD8^+^ T cells per mg heart tissue (*n* = 4 per group). Black dots indicate individual samples.

**Figure 6 ijms-23-07533-f006:**
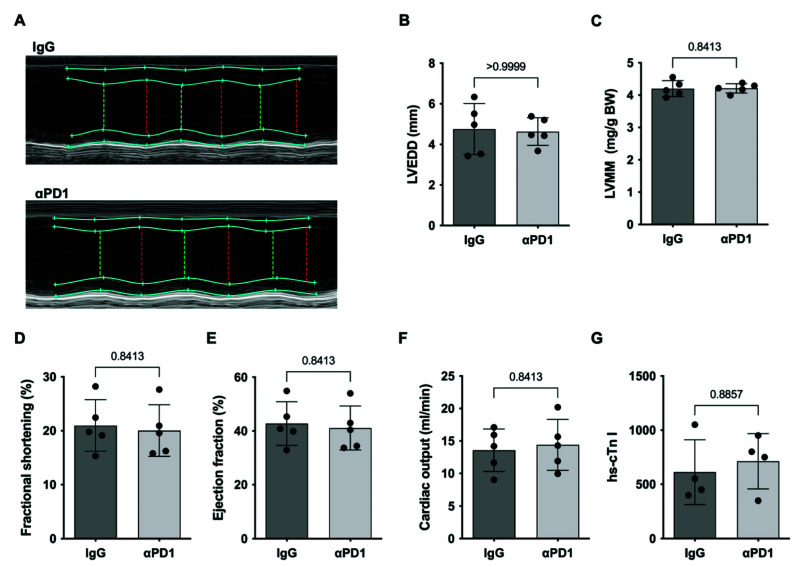
Cardiac function in the early phase. (**A**) Exemplary M-mode images of mouse echocardiography in mice treated with anti-PD1 antibody or IgG control followed by repAMI with 3 days reperfusion followed by quantification of (**B**) left ventricular end-diastolic diameter (LVEDD), (**C**) left ventricular muscle mass index (LVMM), (**D**) fractional shortening, (**E**) ejection fraction, and (**F**) cardiac output. (**G**) show plasma analyses of high-sensitivity cardiac troponin I (*n* = 5 per group). Black dots indicate individual samples.

## Data Availability

The data presented in this study are available on request from the corresponding author.

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
