# Peer review of "PD1 Deficiency Modifies Cardiac Immunity during Baseline Conditions and in Reperfused Acute Myocardial Infarction"

_ijms, 2022, doi:10.3390/ijms23147533_

Round 1
Reviewer 1 Report
The current study investigated the role of PD1/PDL1 in cardiac tissue both in baseline conditions and after ischemia/reperfusion damage. The increasing association between anti-PD1/PDL1 therapies and cardiovascular diseases is an intriguing and promising research topic that needs to be extensively characterized. Thus, the manuscript is quite interesting and well-written. However, I have some major comments that I would like to be addressed:
1. In figure 1, the authors showed some western blot quantifications of target proteins involved in the inflammatory response. However, the quality of the images presented is relatively low. Is it possible to modify them to show clearer signals?
2. When evaluating ERK1/ERK2, I believe it is more important to measure the levels of activation (phosphorylation) of these proteins instead of their expression. The authors should perform a WB testing the phosphorylated form of the proteins and then calculate the ratio between the phosphorylated results and the total levels of expression.
3. In figure 1D, they analyzed the expression of NOX1, concluding that there are no significant differences between wt and Pdcd1 null mice. However, looking at the picture reported, it seems that the expression in wt animals is higher. Could the author double-check this information?
4. The in vivo data shown in Figure 1 gives an overall picture of the cardiac inflammatory status of Pdcd1 null mice at baseline. More detailed information about the expression of these inflammatory molecules in the different cellular populations would provide valuable insights into these mechanisms.
5. The FACS analysis shown in Figure 2 should be performed after dividing the infarct/border zone from the ventricular healthy tissue.
6. Figure 3. Is it possible to perform immunostaining of PDL1 and a marker for cardiomyocytes (i.e. TNNT2) after inducing the ischemia/reperfusion injury?
7. The authors evaluated the extent of the damage after 24 hours of reperfusion (Figure 4) following the treatment with anti-PD1 ICI therapy. However, the assessment of the infiltrating immune cells after repAMI has been done after 72 hours. Is it possible to evaluate the cardiac damage at the same time point?
8. I recommend performing only non-parametric statistical analysis tests because the number of replicates is too low to define a normal data distribution. I suggest using the Mann-Whitney test instead of the T-test and the Kruskal-Wallis test instead of the 1-way ANOVA.
Minor points:
- The authors employed isoflurane to anesthetize the animals but it is well known that this molecule has protective cardiac effects. It is recommended to use medetomidine/ketamine.
- Check the bibliography because the numbers are repeated.
Reviewer 2 Report
This study shows that there are immunological changes due to PD1 deficiency in healthy hearts and in reperfused acute myocardial infarction (repAMI). PD1-deficient mice have increased expression of inflammatory cytokines such as cardiac interleukins and ERK1/2. Mouse in vivo repAMI model and the imaging analysis showed decreased expression of the ligand PDL1 in the infarct region. Pharmacological depletion of PD1 prior to epAMI also increased the number of CD8+ T cells. These results indicate that there is a risk of adverse events with immune checkpoint inhibitor therapy and acute myocardial infarction.
Interesting findings are presented here, but further verification would be needed to prove them.
Comments
What does the number of samples (4 or 3) shown in Figure 1 indicate? To be clearly stated. Also, indicate how it was quantified.
The molecular weight of IL-4 is 12-20kDa. Is the band that appears at 40kDa IL-4?
What is the location of the heart tissue used? In Figure 2 it is left ventricular tissue, is Figure 1 the same? Is it different?
Figure 3 should clearly illustrate the infarcted area with a notation of the tissue site.
In Figure 3, isn't immunostaining necessary before starting repAMI?
How do the authors think the CD8+ T cells that showed an increase are acting?
Line 312; Space between 1:5000Thermo.
Round 2
Reviewer 1 Report
The authors did a great job and significantly improved the quality of their manuscript. They fulfill all the requests.
Reviewer 2 Report
The authors responded to all of my comments satisfactorily, and they modified the manuscript accordingly. I have no further comments.